# Maize adaptation across temperate climates was obtained via expression of two florigen genes

Sara Castelletti[1☯¤a], Aude Coupel-Ledru[2☯¤b], Italo Granato[2], Carine Palaffre[3], Llorenç Cabrera-Bosquet[2], Chiara Tonelli[1], Stéphane D. Nicolas[4], François Tardieu[2], Claude Welcker[2], Lucio Conti[1]*

**1** Department of Biosciences, University of Milan, Milan, Italy, **2** LEPSE, INRAe, Univ. Montpellier, SupAgro, Montpellier, France, **3** Unité Expérimentale du Maïs, INRAe, Univ. Bordeaux, Saint Martin de Hinx, France, **4** GQE—Le Moulon, INRAe, Université Paris-Saclay, CNRS, AgroParisTech, Gif-sur-Yvette, France

☯ These authors contributed equally to this work.
¤a Current address: Transactiva Srl, Parco Scientifico e Tecnologico "L. Danieli", Udine Italy
¤b Current address: University of Bristol, School of Biological Sciences, Bristol, United Kingdom
* Lucio.Conti@unimi.it

**Data Availability Statement:** Genotyping data of the panel used in this study are publicly available at this site: https://data.inra.fr/dataset.xhtml?persistentId=doi:10.15454/GAHEU0. Raw data and

## Abstract

Expansion of the maize growing area was central for food security in temperate regions. In addition to the suppression of the short-day requirement for floral induction, it required breeding for a large range of flowering time that compensates the effect of South-North gradients of temperatures. Here we show the role of a novel florigen gene, *ZCN12*, in the latter adaptation in cooperation with *ZCN8*. Strong eQTLs of *ZCN8* and *ZCN12*, measured in 327 maize lines, accounted for most of the genetic variance of flowering time in platform and field experiments. *ZCN12* had a strong effect on flowering time of transgenic *Arabidopsis thaliana* plants; a path analysis showed that it directly affected maize flowering time together with *ZCN8*. The allelic composition at *ZCN* QTLs showed clear signs of selection by breeders. This suggests that florigens played a central role in ensuring a large range of flowering time, necessary for adaptation to temperate areas.

## Author summary

The cultivation of maize in temperate climates required the suppression of photoperiod sensitivity and the selection of a wide range of flowering time for the adaptation to local environmental constraints. Photoperiodic flowering requires the production of a systemic protein signal referred to as the florigen, which is transcriptionally activated in leaves. A key question is to what extent the expression of the florigen gene can explain flowering time variability of temperate maize, where photoperiod sensitivity has been eliminated by artificial selection. Our results reveal large variability in two related florigen genes which is strongly correlated with variability in flowering time. Using association genetics approaches we could detect genomic regions responsible for the expression of the two florigens that precisely co-localise with flowering time-related regions, thereby supporting

BLUES of gene expression and phenotypes as well as the description of the lines used are publicly available at this site: https://data.inra.fr/dataset.xhtml?persistentId=doi:10.15454/XTHN7I Additional relevant data are within the manuscript and its Supporting Information files.

**Funding:** This study was funded through FLORIMAIZE "Role of florigen proteins in maize developmental reprogramming under drought stress" Project (FC ID 2013-1889; AF ID 1301-006) jointly supported by Agropolis Fondation (through the "Investissements d'avenir" programme with reference number ANR-10-LABX-0001-01") and Fondazione Cariplo (for SC, ACL, IG, CP, LCB, CT, FT, CW, LC). http://www.fondazionecariplo.it/it/progetti/ricerca/bando-congiunto-fondazione-cariplo-agropolis-fondation-ceres.html. Genotyping data used for the project were assembled within Amaizing project (French National Agency ANR-10-BTBR-01) (for LCB, IG, SN, FT, CW). https://anr.fr/ProjetIA-10-BTBR-0001. Support for the phenotyping platform derived from the Projects PHENOME Emphasis (ANR-11-INBS-0012) (for LCB, FT, CW). https://anr.fr/ProjetIA-11-INBS-0012. The funders had no role in study design, data collection and analysis, decision to publish, or preparation of the manuscript.

**Competing interests:** The authors have declared that no competing interests exist.

the significance of quantitative changes in florigen levels in driving flowering time variability. Markers associated with florigen expression/flowering time display significant signatures of selection indicating that variable patterns of florigen accumulation underpin the adaptation of temperate maize flowering.

## Introduction

Maize was domesticated 9000 years ago in central Mexico from teosinte (*Zea mays* subsp. *parviglumis*), a tropical plant that only flowers under short days [1]. In contrast to its wild ancestor, modern maize shows a large geographic distribution from 0 to 50˚ N. Adaptation to temperate latitudes, with long days during summer, required breeders to first suppress the short-day requirement for floral induction, but a second adaptation was also required. Because the time to flowering strongly depends on temperature, it is shorter for a given non-photoperiodic genotype at lower than at higher latitudes of the temperate area. A long intrinsic cycle duration is necessary to counteract this effect in warmest areas, thereby avoiding decreases in cumulated photosynthesis and biomass. Conversely, a short intrinsic duration is needed in the coolest areas to avoid grain filling to occur in Autumn with low light and temperature [2]. Accordingly, the range of time to flowering in current germplasm typically ranges from 35 to 90 days after sowing in a given site [3].

About ninety quantitative trait loci (QTLs) in the maize genome account for the natural diversity in flowering time [4–8] and most of them explain a small proportion of the phenotypic variance, with prevalently additive effects [4]. The molecular characterization of a few large-effect alleles helped uncover key events underpinning the flowering adaptation of maize from tropical to temperate climates [9]. These alleles were found to regulate the expression of florigens, i.e. floral genes expressed in the leaf vasculature. Florigen proteins are members of the Phosphatidyl Ethanolamine Binding Proteins (PEBPs) family that include FLOWERING LOCUS T in *Arabidopsis* and Heading date 3a in rice that move systemically to promote flowering at the shoot apex [10]. The maize genome encodes several florigen genes, including *ZEA CENTRORADIALIS 8* (*ZCN8*) that acts as major florigen [9,11–13]. Its expression only occurs under short-day conditions in tropically adapted maize, but not in long days [13]. The uncoupling of *ZCN8* expression from short day-dependent activation occurred in a stepwise manner through the selection of mutations in the *ZCN8* expression regulatory processes. For example, independent transposon insertions at two related *ZmCCT* genes (*9* and *10*) reduce maize flowering photoperiod sensitivity [14,15]. The active (expressed) alleles of *ZmCCT9* and *10* of tropical maize delay flowering under long days by repressing *ZCN8* [13,16]. The gradual northbound expansion of maize was also driven by variations at cis-regulatory sequences of the *ZCN8* promoter [9], which modify the binding landscape for different transcription factors [17]. Another *ZCN8* regulatory node involves *ZmMADS69*, which in turn represses the *ZCN8* negative regulator *Related to apetala2.7* (*Rap2.7*) [18]. Both *ZmMADS69* and *Rap2.7* map within well-characterized QTLs of flowering time suggesting that allelic variations at these genes control part of the diversity of flowering time of temperate maize [19–22]. Classic mutagenesis screens also identified major genes that affect flowering time, thereby helping to refine the *ZCN8* regulatory pathway with additional components. These include *Indeterminate1* (*Id1*) and *delayed flowering 1* (*dlf1*) that encode, respectively, an upstream regulator required for *ZCN8* expression [23,24] and a bZIP transcription factor presumably required for ZCN8 protein signalling at the shoot apex [13,25].

Although the above studies reveal a prominent role for *ZCN8* activation in the control of flowering time in short vs long days, its contribution to the genetic variability of flowering time in non-photoperiodic temperate maize varieties is less clear. Indeed, it has been argued that the variability in flowering time in temperate maize is controlled by other pathways that act in parallel to or downstream of *ZCN8* transcriptional activation [26]. Two elements challenge this view: (i) the downregulation of *ZCN8* in temperate maize leads to a significant delay in flowering time (albeit less prominent compared to *id1* or *dlf1* mutants) [13,16,18], (ii) QTL mapping shows that flowering time is linked to allelic variations at a genomic region that harbours *ZCN8* (also known as the *Vgt2* QTL) [6,19,27,28].

A genome-wide study of temperate maize germplasm was necessary to reconcile these views on the potential role of florigen expression in the control of flowering time of temperate maize. Here we address this question by jointly measuring flowering time and the expression of *ZCN8*, and of two related genes with unknown variability, *ZCN7* and *ZCN12* [12], in a panel of 327 maize temperate lines. Our study revealed large variations in *ZCN8* and *ZCN12* transcript levels, which closely correlate to flowering time. Regulatory regions for *ZCN8* and *ZCN12* coincided with flowering time QTLs and we detected signatures of selection for some of these regions. Collectively, these results suggest that the adaptation of maize within temperate regions derives from the control of florigen pathways.

## Results

### *ZCN8* levels highly correlate with flowering time in temperate maize

We first measured the time from germination to pollen shedding on florets (anthesis), to the emergence of silks from the ear (silking) and also the final leaf number which reflects the duration from germination to floral transition, in four experiments in a phenotyping platform and in the field. Large variability was observed in two platform experiments for anthesis (BLUES ranging from 56 to 99 $d_{20°C}$), silking (from 57 to 101 $d_{20°C}$) and leaf number (from 13 to 25) (S1A and S1B Fig), resulting in narrow-sense heritability up to 0.80 (S1 Table). Field experiments provided similar results to those obtained in the platform (Fig 1A, S1C Fig, S1 Table).

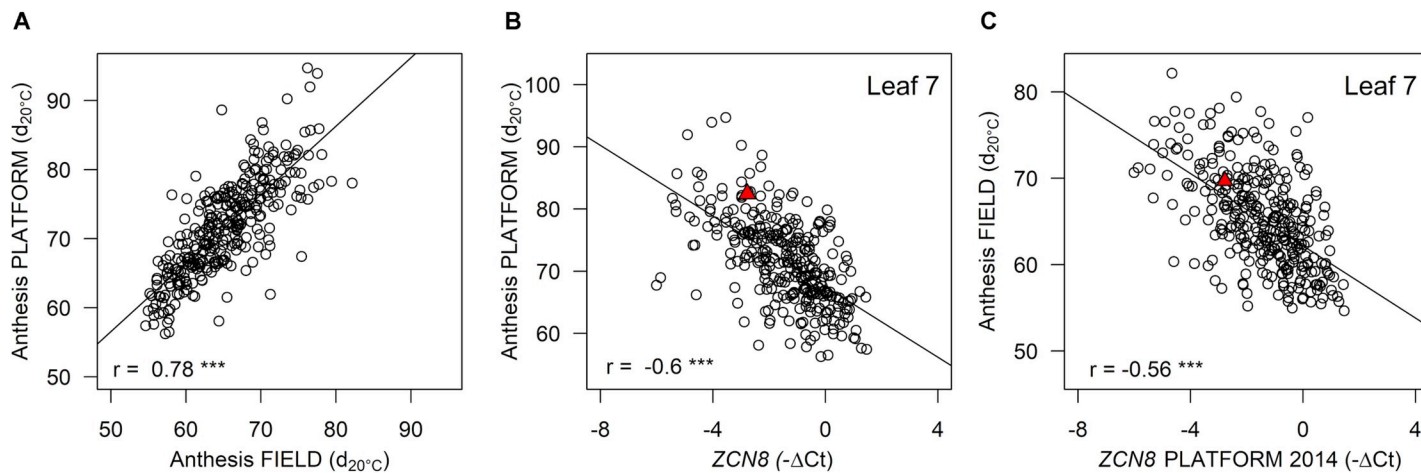

**Fig 1. *ZCN8* contributes to flowering time variability in temperate maize lines.** (A) Relationship between genotypic values for anthesis in field (2015–2016) and platform (2014) experiments. (B) Anthesis time plotted against *ZCN8* expression after the floral transition (platform experiment in 2014). (C) Relationship between anthesis measured in the field (2015–2016) and *ZCN8* expression in the platform experiment (2014). In (A), (B) and (C) *n* = 313 lines. Genotypic values are BLUEs. Anthesis time expressed in equivalent days at 20°C after germination. The red triangle represents genotypic values for the reference line B73. Correlation coefficients are displayed together with their significance (*** p < 10$^{-4}$).

We then measured florigen expression of the same lines, in samples collected from fully expanded leaves of plants grown in the phenotyping platform. Samples were collected in the early morning at two dates, when leaves 4 and 7 reached ligulation, representing *bona fide* pre and post floral transition [13]. Care was taken to precisely record the harvest time of each individual sample and to record the developmental stage of every sampled plant, which slightly differed between plants because of the genetic variability of the phyllochron [29]. Mixed models including a fixed effect of sampling date and time of day (both of them small, S2A–S2D Fig), together with spatial effects in the greenhouse, were used to extract genotypic values for each line (S3C and S3D Fig).

The *ZCN8* expression displayed large genetic variability. The four biological replicates, analysed at two sampling dates, provided highly correlated expression levels (up to r = 0.88) when values were normalised by those of the housekeeping genes *UBIQUITIN-CONJUGATING ENZYME E2* (*UCe*) and *ELONGATION FACTOR 1 ALPHA* (*EF1a*) (S3A and S3B Fig). *ZCN8* accumulation significantly increased between the two sampling dates (S4B Fig), consistent with previous studies [13], with higher heritability in the second than in the first date ($h^2$ = 0.64 and 0.58, respectively). Still, we could detect a significant correlation in *ZCN8* accumulation between the two sampling dates (r = 0.53, S4A Fig). We thus focused on *ZCN8* expression detected at the second sampling date (leaf 7 stage).

Anthesis time was significantly correlated with the BLUEs of *ZCN8* (r = -0.6, Fig 1B) and so was the total leaf number (r = -0.65, S5A Fig). The ranking of genotypes for *ZCN8* was confirmed in an independent platform experiment (year 2015) on a subset of genotypes (n = 4 replicates for 276 lines, r = 0.63, S5C Fig). *ZCN8* transcript levels detected in the phenotyping platform were also highly correlated with flowering times measured in field experiments (Fig 1C and S5A and S5B Fig). For any given expression level at the second sampling date, lines characterised by a strong increase in expression flowered earlier than the other lines (r = -0.36, S4C Fig). This suggests that constitutively high levels of *ZCN8* and an increase in *ZCN8* expression during development both contribute to early flowering time in temperate maize.

## ZCN12 is a novel regulator of maize flowering and it is co-expressed with ZCN8

Although *ZCN12* levels have not been documented so far for their role in flowering time, they showed high heritability (narrow-sense $h^2$ = 0.67, S1 Table) and displayed a strong correlation with anthesis time (r = -0.64, Fig 2A) and leaf number in all platform and field experiments (S6A Fig). *ZCN12* expression also tightly correlated with that of *ZCN8* (r = 0.76, Fig 2B and S6B Fig), suggesting common regulatory mechanisms. This was in contrast to the *ZCN7* expression that displayed low heritability (narrow sense $h^2$ = 0.11, S1 Table) and a weaker correlation with flowering time (r = -0.31, S6C Fig) than *ZCN8*, despite synteny [30] and shared biochemical functions [31]. *ZCN7* and *ZCN8* expressions were also weakly correlated (S6C Fig). The accumulation of *ZmCONZ1*, a putative upstream regulator of the florigen genes in maize [32], was not linked to *ZCN8* or *ZCN7* expressions, nor with flowering time (S6D Fig).

In order to demonstrate the role of *ZCN12* on flowering time, we cloned the *ZCN12* gene for over-expression studies in *Arabidopsis thaliana*. This demonstrated a strong florigenic activity for the ZCN12 protein. Transgenic lines overexpressing *ZCN12* flowered significantly earlier compared with control plants (Fig 2C–2E). Sixteen of the 40 transgenic plants showed an extremely early flowering phenotype characterised by the conversion of the shoot apical meristem in a terminal flower, with no observable vegetative leaves. The same transformation for the *ZCN8* gene also caused significantly earlier flowering, but with only two cases of extreme early flowering.

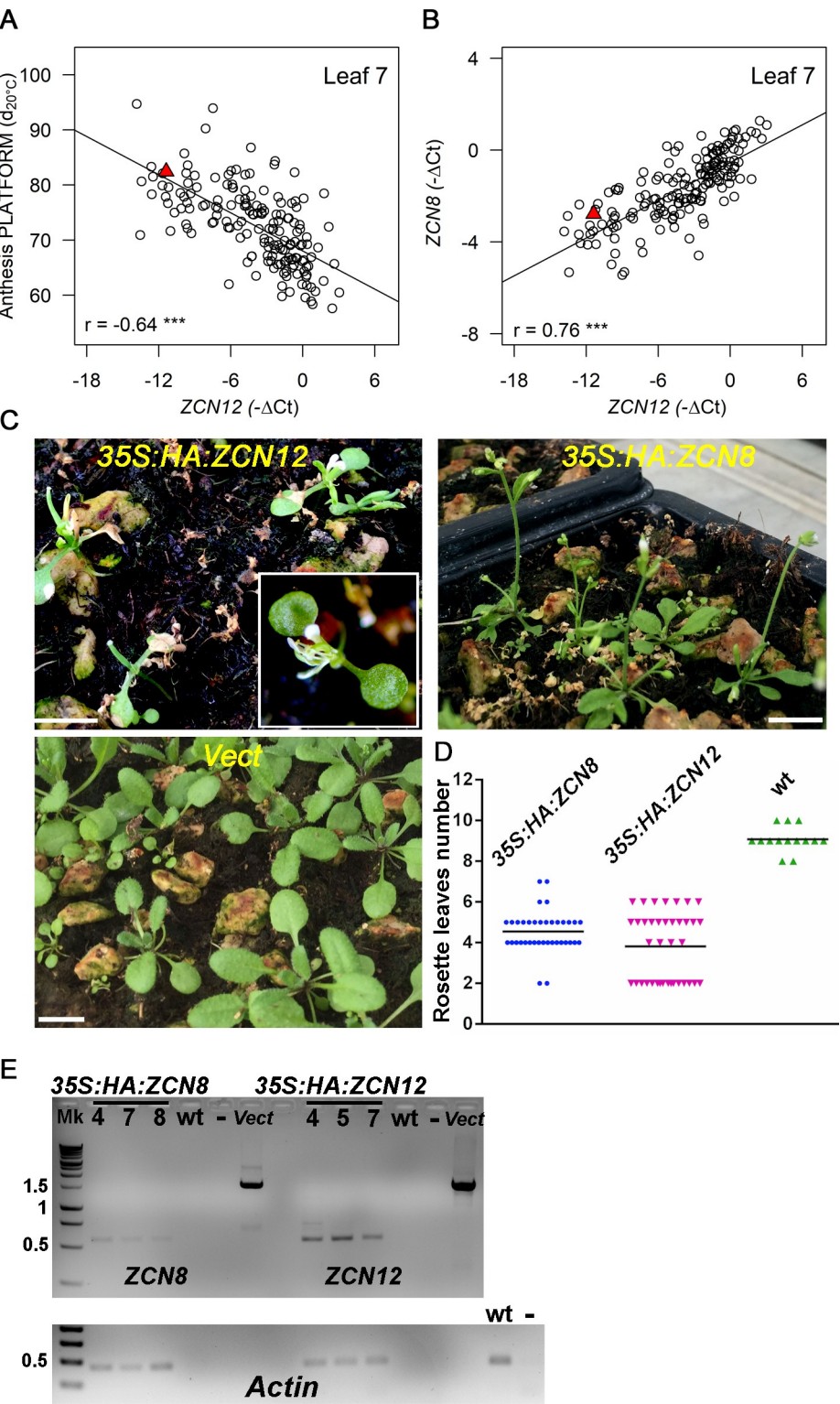

**Fig 2. ZCN12 is a novel maize florigen co-regulated with ZCN8.** *ZCN12* expression correlates with anthesis time (platform experiment 2014), (A), and with *ZCN8* expression (B). Genotypic values are BLUEs. Pearson's correlation coefficients are indicated with their significance (***, $p < 10^{-4}$). The red triangle represents genotypic values for the reference line B73. In (A) and (B), n = 173. (C) Arabidopsis T1 plants overexpressing *ZCN12* but not *ZCN8* show an extreme early flowering phenotype compared to empty vector transformation (Vect.); plants were photographed 3 weeks after sowing, horizontal bar, 1 cm. (D) *ZCN8* and *ZCN12* T1 over-expressing plants are earlier flowering than wild type

(p < 0.01, one-way ANOVA with post-hoc Tukey HSD test). Horizontal bars, mean values of rosette leaves number. n = 37, 38 and 14 for, respectively, *35S:HA:ZCN8 35S:HA:ZCN12* and wild type (empty vector). (E) RT-PCR analysis on *ZCN8/12* transcripts derived from independent T2 transgenic plants of the indicated genotypes. Amplification of vectors (Vect) harbouring genomic versions of *ZCN8* and *ZCN12* afforded a positive control whereas untransformed wild type and no DNA reactions (-) were used as negative controls. Actin expression was used for normalization (lower panel). Numbers on the left are kb based on DNA ladder (Mk) migration. Amplifications were conducted for 25 cycles.

## Direct roles of *ZCN12* and *ZCN8* in the control of flowering time

We then showed that *ZCN12* had a direct effect on maize flowering time. Indeed, structural equation modelling rejected models in which *ZCN12* had an upstream effect by controlling *ZCN8* and/or *ZCN7* expressions, which would in turn control anthesis (Fig 3A and 3B, S7A and S7B Fig). Conversely, our analysis supported models in which *ZCN12* accumulation had a direct effect on anthesis time, with either *ZCN12* being controlled by *ZCN8*, or *ZCN8* being controlled by *ZCN12* or co-variation of both genes (Fig 3C, S7C Fig). The latter models displayed good indices of fit (RMSEA < 0.05, comparative fit index > 0.95). They were considered as equally plausible, thereby supporting the existence of a two-tier florigen system in maize in which *ZCN12* acts in parallel with *ZCN8*.

## *ZCN8* and *ZCN12* regulators map to region controlling maize flowering time

We then analysed to which extent genomic regions controlling florigens expression co-localised with regions controlling flowering time. A genome-wide association study (GWAS) identified 570 QTLs which largely overlapped between field and greenhouse experiments (S2 and S3 Tables), resulting in 214 meta-QTLs that harboured at least one QTL (Fig 4A) of flowering time, in either field or greenhouse experiments. An eQTL GWAS detected 52 meta-QTLs for *ZCN8*, among which half co-localised with flowering time QTLs. We also detected 85 meta-

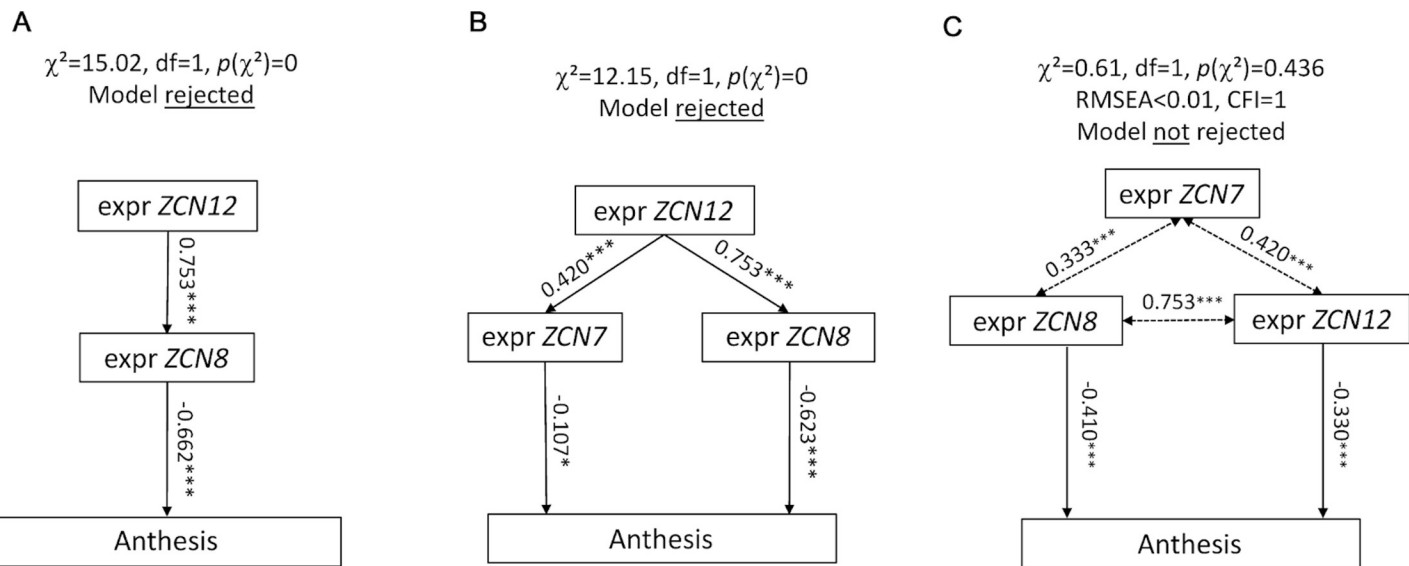

**Fig 3. *ZCN12* directly affects anthesis.** Three families of path diagrams were tested against the whole dataset (BLUEs, platform experiment of 2014). Models in which *ZCN12* has an indirect effect on anthesis (A and B) were rejected, whereas models in which both *ZCN8* and *ZCN12* directly affect anthesis were accepted, with an indirect effect of *ZCN7* (C). Arrows, linear functional relationships between anthesis and florigen expression. Simple-headed arrows, relationships are considered as causal by the model. Double-headed arrows, free correlations. Standardized path coefficients are indicated on each arrow with the level of significance (***: P < 0.001, ns: not significant). Other models tested are displayed in S7 Fig.

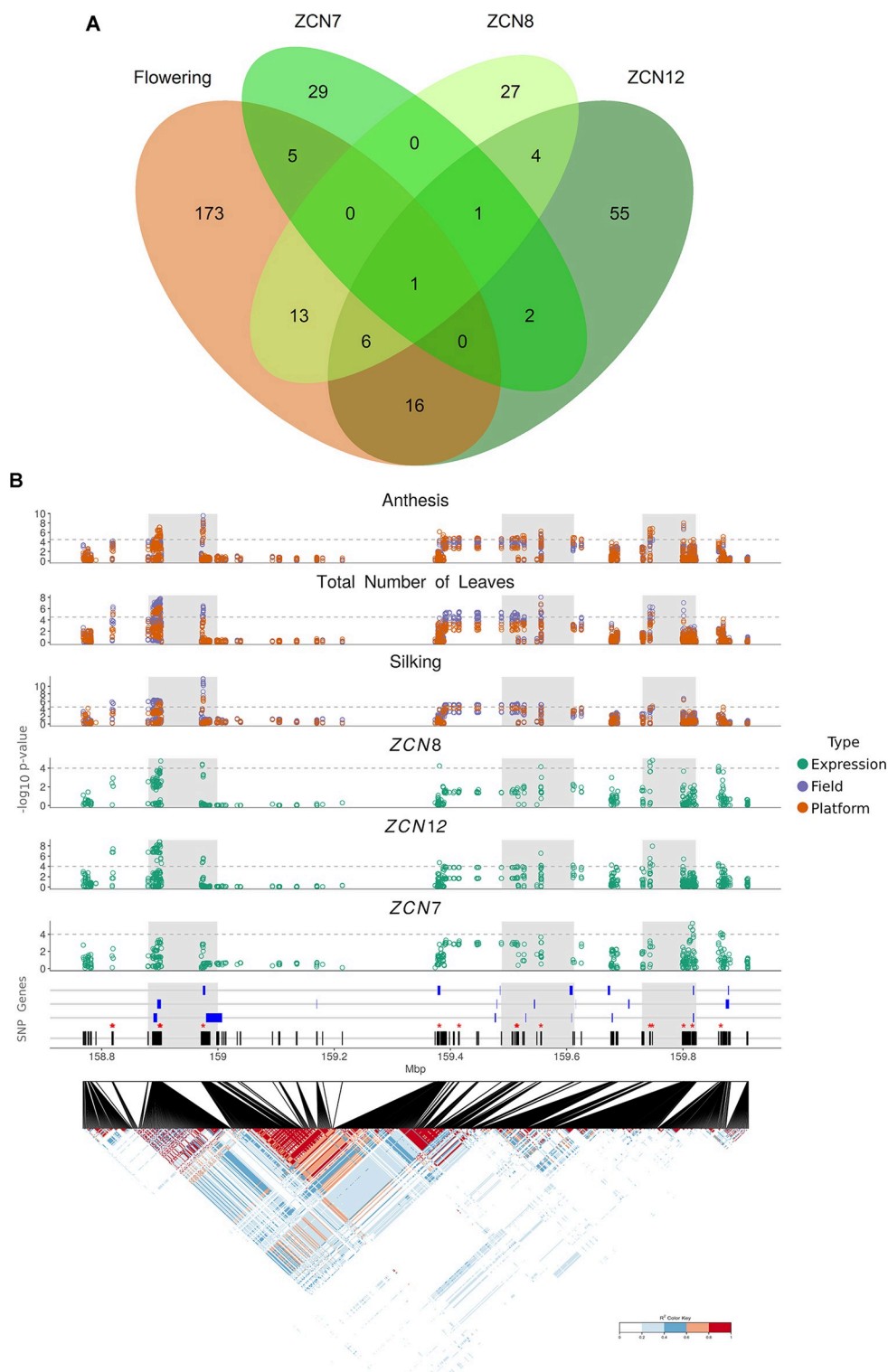

**Fig 4. *ZCN* eQTLs are major contributors to the genetic variance of flowering time.** (A) *ZCN*-related QTLs overlap with flowering time QTLs. Venn diagram highlights 'meta-QTLs' shared among flowering-related traits and *ZCN* transcripts accumulation. "Flowering" meta-QTLs are defined as regions containing at least one QTL among anthesis, silking, total number of leaves in at least one experiment (platform, field, years). "*ZCN-*" meta-QTLs are defined as regions containing at least one *ZCN-* eQTL in at least one experiment (platform 2014–2015) and leaf stage. (B)

Regional association plot for one meta-QTL on chromosome 3. Distribution of the -log$_{10}$(pval) for all variants in the region. The dotted grey line corresponds to–log$_{10}$(pval) = 4.5 for anthesis-related traits evaluated in the field (purple) and platform (orange) and 4 for the expression traits evaluated in the platform (green). (C), linkage disequilibrium (LD) heat map of all SNPs in the QTL showing the local LD (r$^2$) between all the variants; Black lines represent the distribution of the SNPs and the blue boxes represents the genes mapped for the region. Red asterisks represent the position of the most significant SNPs. (See also S4 Table for candidate genes). Grey areas represent sub-regions harbouring the most significant SNPs and genes and having low LD between them on average.

QTLs for *ZCN12* and 38 for *ZCN7* (Fig 4A). 12 meta-QTLs were shared between *ZCN8* and *ZCN12*, of which 7 mapped to flowering time QTLs (Fig 4A). They also overlapped with flowering time QTLs detected in published studies [7] (S4 Table). *ZCN* eQTLs had a high contribution to the genetic variance of flowering time as shown by mixed-models applied to individual experiments (S5 Table). Indeed, in the platform experiment of 2014, the 19 *ZCN* eQTLs that co-localised with anthesis QTLs accounted for 57% of the genetic variance of anthesis in the platform experiment (vs 75% for all 43 QTLs of anthesis). Similar results were found when considering anthesis in the field, or total leaf number in each experiment (S5 Table).

We inspected the *ZCN* eQTLs for the presence of candidate genes or QTLs already known to control flowering time. In the Bin 3.05 hotspot region known for flowering time regulation [21,33], we identified five *ZCN*-related eQTLs which co-localised with at least one QTL of flowering time (S3 Table). An eQTL of *ZCN8/12* (158.73–159.93Mb) was located at a genomic position compatible with the newly described transcriptional regulator *ZmMADS69* (*GRMZM 2G171650*) [18,21,34] (Fig 4B and 4C). Another eQTL region (Bin 8.05) comprised the gene *GRMZM2G700665* (also known as *Rap2.7*) (S8B Fig and S4 Table), encoding a floral repressor [19] and negative regulator of *ZCN8* [18].

Analysis of published whole-genome expression datasets [35] was used to check if transcripts whose expression correlated with *ZCN8* were over-represented at *ZCN* eQTLs positions. We confirmed significant correlations between anthesis and *ZCN8/12/7* expression (n = 181 lines, r = up to - 0.54) (S9A–S9C Fig). Furthermore, these datasets revealed strong patterns of co-regulation not only between *ZCN8* and *ZCN12* but also between *ZCN8* and *ZCN7* (n = 204 lines, r = up to 0.76) (S9D–S9F Fig). Additional 549 transcripts showed a significant correlation with *ZCN8* expression levels (S6 Table). These were parsed in two clusters composed of genes with opposite trends to *ZCN8* gene expression and correspondingly opposite association with anthesis (S9G Fig). Neither *ZmMADS69* nor *Rap2.7* nor any other gene in the eQTLs regions within Bin 3.05 were co-expressed with *ZCN8*, and co-expressed transcripts were not particularly enriched in eQTLs regions (S4 Table).

## Signs of selection at *ZCN8* and *ZCN12* eQTLs to control maize flowering time

The genetic groups in the studied panel showed markedly different values of *ZCNs* expression levels and anthesis dates (S10A,S10B, S10D and S10E Fig), and clear co-expression of *ZCN8* and *ZCN12* within each group (S10C and S10F Fig). The observed differences in flowering time between admixture groups may be due to selection by past breeding of temperate maize lines. Consistent with this hypothesis, we found significant differences, between the seven genetic groups, in allelic distribution at QTLs showing co-location between *ZCN8/12* expression and flowering time (Fig 5A). Alleles for increased expression of *ZCN8* and *ZCN12* were more frequent in genetic groups bred for early maturity (Fig 5A). Hence, a pool of early maturity alleles was represented within all genetic groups, but in different proportions, suggesting that some combinations were preferentially selected in different groups.

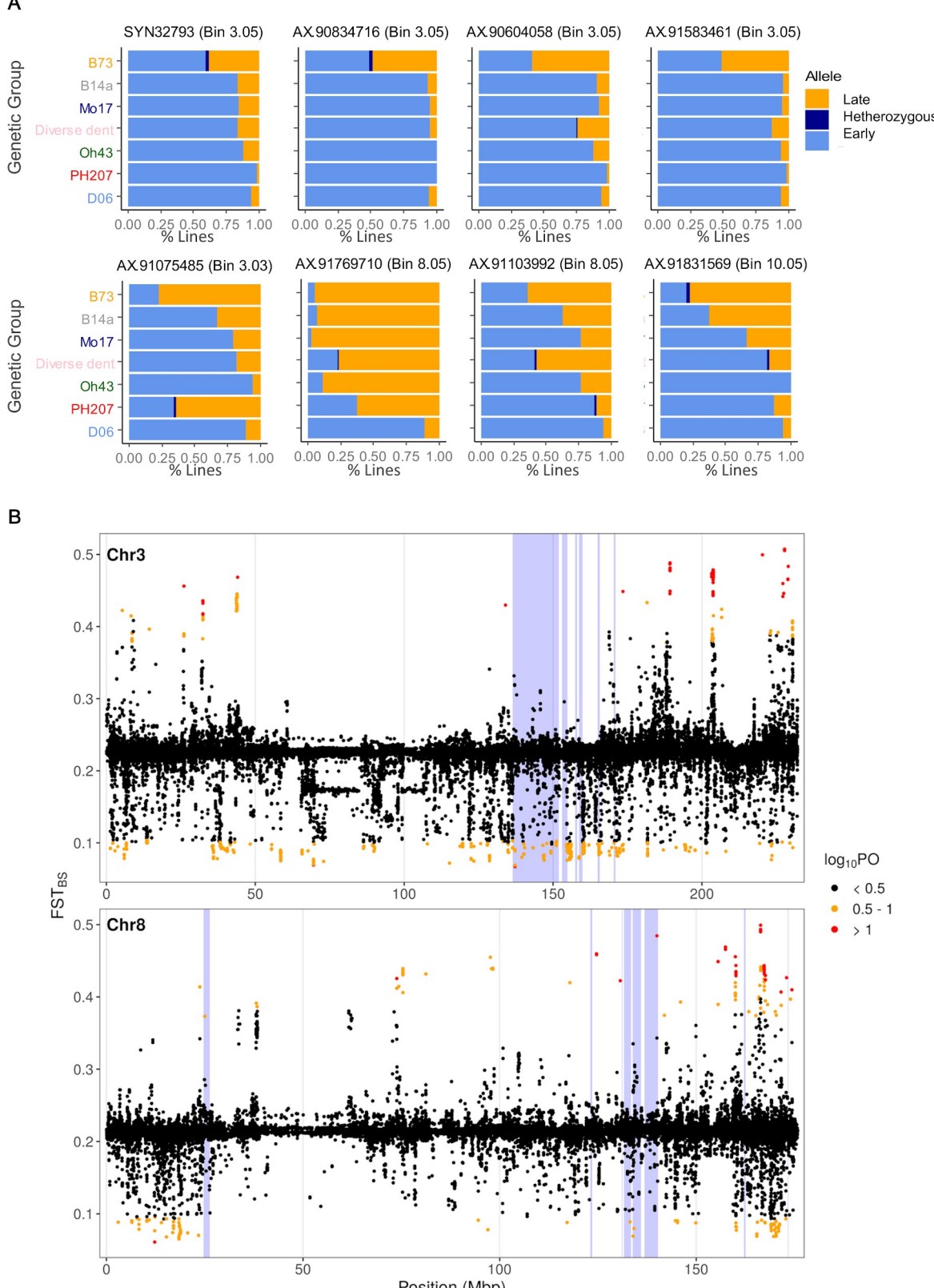

**Fig 5. Flowering time precocity of temperate maize derives from selection at _ZCN8/12_ regulatory regions.** (A) Allelic distributions within genetic groups for 8 major QTLs. The SNPs displayed belong to the core set of 19 QTLs common to anthesis in the platform

experiment in 2014, and *ZCN* transcript accumulations. At each locus the orange colour corresponds to the 'late allele' (i.e. conferring delayed anthesis), which is for all 8 SNPs the B73 allele. Genetic groups are listed according to the flowering time phenology (late to early, top to bottom). (B) Genome-wide distribution of FST Bayescan (FST$_{BS}$) along chromosome 3 and 8. SNPs under selection are highlighted in different colors to reflect the different level of confidence for selection (Black: low confidence, Orange: substantial, Red: strong). Regions shaded in blue correspond to meta-QTLs intervals.

We further tested for genome-wide deviation from neutrality in our panel for the chromosomes 3 and 8 harbouring major meta-QTLs. F$_{st}$ outliers approach using BayeScan highlighted SNPs under selection in the *ZCN*-related QTLs and other regions that might contribute to flowering time ([Fig 5B] and [S7 Table]). When comparing all six populations (excluding the admixture group 'diverse dent'), the BayeScan revealed SNPs under selection ranging from a $log_{10}$(PO)—posterior odds—of 0.5 to 1 (substantially significant) to higher than 1 (highly significant). Most importantly, this analysis revealed SNPs under selection comprised in the *ZCN*-related QTLs regions, further indicating the contribution of *ZCN8/12* regulation to flowering variability.

## Discussion

### *ZCN12* and *ZCN8* are co-regulated by trans-acting factors to control temperate maize flowering

Our study provides an example for how shifts in reproductive development that enable local adaptation can emerge through different florigen levels. Besides confirming the role of *ZCN8* in flowering time regulation we also uncover the role of *ZCN12* in this process. Interestingly *ZCN12* is more distantly related to *ZCN8* than *ZCN7* [12] and located in a region which is not syntenic to *ZCN8* [30], yet it underwent a similar regulatory trajectory to *ZCN8* as shown by the strong co-variation across different lines and experiments.

GWAS approaches support the major contribution of trans-acting factors in driving the expression of *ZCN8* and *ZCN12* in temperate maize. We also detected a SNP associated with anthesis and silking located approximately 1.7 kbp upstream of *ZCN8* (i.e. AX-91100620, [S2 Table]). However, this and eight other SNPs located in this putative regulatory region were not associated with *ZCN8* transcript levels. It is unlikely that the absence of cis-type eQTLs for *ZCN8* could depend on insufficient coverage of the genetic diversity, since they were discovered on a large set of diverse inbred lines: 32,758 accessions for 6 GBS markers, [36] and 30 accessions for 3 Affymetrix Axiom markers [34]. Multiple trans-acting genes and complex long-range regulatory interactions are thus more likely responsible for the accumulation of *ZCN8/ZCN12*. Supporting this hypothesis, we detected trans-eQTLs for *ZCN8/ZCN12* that colocalize with two genomic regions—*Vgt1* and *Vgt3*—carrying genes involved in flowering time variation ([Fig 4B], [S8B Fig]). These regions include transcriptional regulators known to be under selection in maize latitudinal adaptation, namely *ZmMADS69/Vgt3* [18,21,33] and *Rap2.7/Vgt1* [19,22]. We did not detect any obvious candidate genes associated with photoperiodic signalling (e.g. *ZmCCT10*), suggesting its small contribution in *ZCN8/12* regulation in our panel.

*ZmMADS69* and *Rap2.7* did not show significant trends of co-expression with their trans-regulated targets *ZCN8/12*. It is thus possible that that there are many more genes of smaller effect contributing to *ZCN* variations which were not captured in our GWAS. These loci could affect the activity or accumulation of some of the abovementioned transcription factors. For example, *microRNA172s* are known to interfere with the translation of *APETALA2* genes [37] (such as *Rap2.7*), indicating that transcript accumulation of *Rap2.7* may not reflect its gene product levels. Another not mutually exclusive explanation may be that regulatory networks that impact on *ZCNs* levels are transient or highly tissue-specific, which would make it difficult to compare independent expression datasets.

We found limited numbers of genes located within eQTLs regions and co-expressed with *ZCN8* and *12* (S4 Table). These included a putative RNA binding protein *GRMZM2G107491* (underlying a *ZCN12*-specific eQTL), and two chloroplast-associated proteins (underlying different *ZCN8* eQTLs), implying carbon metabolism in *ZCN8* regulation. Carbon status is linked to the transcriptional regulation of the *Arabidopsis* florigen *FLOWERING LOCUS T* [38] independent of photoperiod conditions. Moreover, recent studies indicate that temperate maize flowering is associated with expression of genes involved in carbon partitioning (which provides a different molecular signature from photoperiodic flowering induction) [24]. We note that patterns of co-variations may not necessarily define causal, but also indirect relationships. Also, since *ZCN8* levels are linked to flowering, our list of *ZCN8* co-expressed genes would lead to overemphasizing the association between gene expression and flowering. Thus, co-expressed genes might also be associated with a different, more biologically distal phenotype. Overall, our data support the contribution of carbon metabolism upstream of *ZCN8* and provide an initial framework for the identification of photoperiod independent mechanisms that shape *ZCN8* levels across temperate maize varieties.

### *ZCN8/12* levels underpin independent events of selection for flowering adaptation

A large body of evidence highlights the importance of the regulation of gene expression in maize evolution and adaptation [39,40] and the contribution of the expressed genome to the heritable phenotypic variance [41]. It is hypothesised that temperate maize is the product of a northbound expansion through North America which was driven by the accumulation of an increasing number of early flowering alleles [42,43]. *Rap2.7* and many other loci were under selection during the adaptation to high latitudes [22,27] and general patterns of differentiation for flowering time precocity closely followed the proportion of early alleles found in northern flints (earliest), corn belt dents (intermediate) and tropical maize genomes (late maturity). Besides revealing that at least some of these loci exert their effects on flowering time by affecting *ZCNs* levels, our study indicates that diversification in flowering can be imposed by combinatorial allelic effects in a relatively small number of *ZCN*-related eQTLs regions. While flowering time may represent the main trait under selection, these alleles (through their effect on florigen levels) may also affect other phenotypic outputs, for example florigen levels have been related to inflorescence development in other grasses [44]. We finally provide evidence for different signatures of selection in the main regions previously mapped in the genetic control of flowering time, thus confirming their potential contribution in the adaptation process.

### Conclusion

We show that florigen genes play an essential role in the genetic variability of flowering time in those lines in which breeding suppressed photoperiodism. We reveal the role of a novel florigen, *ZCN12*, a non-syntenic paralog of *ZCN8*, whereas other related candidates had no clear role. Our study points to combinations of trans-type regulatory mechanisms underpinning selection for flowering time in different genetic groups through florigen levels.

## Materials and methods

### Plant material

A maize collection of 354 lines was assembled in the frame of the DROPS (European project FP7-244374) and Amaizing projects (French National Agency ANR-10-BTBR-01). It consists of historical and breeding American and European dent lines from the main founder group of

modern hybrids [33]. Lines were genotyped using 50K Infinium HD Illumina array [45], a 600K Axiom Affymetrix array [34] and a set of 500K markers obtained by Genotyping by Sequencing (GBS) [33]. Genotyping data were filtered and missing data were imputed using Beagle V3.3.2 [46] after assembling three sub-matrices and removing duplicate loci based on their physical position. After quality control, 758,863 polymorphic single-nucleotide polymorphisms (SNPs) were retained for GWAS analyses. All physical positions referred to hereafter are based on the B73 reference genome [47] RefGen_V2. Based on the 50K [33], seven genetic groups were identified using the ADMIXTURE software. These groups were constituted by: (i) 53 lines in the Non Stiff Stalk (Iodent) family traced by PH207, (ii) 18 European dent in the one traced by D06, (iii) 38 lines in the Lancaster family traced by Mo17, (iv) 17 in the one traced by Oh43, (v) 39 lines in the stiff stalk family traced by B73, (vi) 43 lines in the one traced by B14a, and (vii) 146 lines that did not fit into the six primary heterotic groups, such as W117, NC358, and F252 (referred to as "diverse dent"). B73 was considered as the reference line [47].

## Assessment of flowering related traits in the field

The panel was assessed in two consecutive field experiments for flowering related traits. Experiments were conducted at Saint Martin de Hinx (INRA experimental station, 43˚N, 1.3˚W, France) in spring-summer 2015 and 2016, using alpha-lattice designs with two replicates under well-watered conditions piloted by soil sensors. Male (anthesis) and female (silking) flowering dates, as well as the total number of leaves (including the ones lost) produced were measured for all the lines. Air temperature was measured every hour in each experiment at 2m height over a reference grass canopy, and the progression of the crop cycle site was characterized via thermal time after emergence, expressed in equivalent days at 20˚C ($d_{20˚C}$) [48].

## Flowering-related traits measurements in the platform

Two subpanels of 327 lines and 276 lines were grown in the Phenoarch platform [49] (https://www6.montpellier.inrae.fr/lepse_eng/M3P) respectively in autumn 2014 and spring 2015 under natural fluctuating daylength conditions (with a minimum daylength of 14 hours). The two subpanels were optimized using the algorithm described in Rincent et al. [50] that allows to maximise genetic information by discarding individuals that might be more easily predictable. Five plants per line were grown under well-watered conditions and daily assessed for developmental traits. The number of visible and ligulated leaves of every plant was scored twice a week. Anthesis, silking (in $d_{20˚C}$) and the total number of leaves produced (including the ones lost) were assessed for all the lines.

## Tissue sampling for transcripts expression analysis

Maize leaves were sampled at two developmental stages, ligulated leaf 4 and 7 (L4 and L7, respectively) from plants growing in the platform. The sampling day was decided based on B73 stage, which was used as a morphological reference. Four plants were sampled for each line. Samples deprived of the midvein were collected in the distal portion of fully expanded leaves (with clearly visible ligule) of each plant at approximately the same time of the day (within 90 minutes from sunrise). Sampling was conducted over three consecutive days for each sampling stage, and the fourth sampling was done soon after the third one. The collected material was kept in liquid nitrogen and the tubes were stored at -80˚C.

## RNA extraction and cDNA synthesis

The frozen samples were ground at high speed with glass beads in a TissueLyser II apparatus (Qiagen). Qiazol reagent (Qiagen) was added to individual tubes and total RNA was extracted using the Direct-zol™-96 kit (Zymo Research); on-column DNase I digestion was performed following the kit protocol. Total RNA was then quantified using a NanoDrop One instrument (ThermoFisher) and 1000 ng of each RNA sample were used for retrotranscription with the High-Capacity cDNA Reverse Transcription Kit (Applied Biosystems). The cDNA samples were diluted 1:5 with water before proceeding to qPCR experiments.

## TAQMAN and SYBR green assays design, implementation and analysis

The genomic and coding sequences of the targets *GRMZM2G179264* (*ZCN8*), *GRMZM2G1 41756* (*ZCN7*), *GRMZM2G103666* (*ZCN12*) and *GRMZM2G405368* (*ZmCONZ1*) and that of reference genes *GRMZM2G102471* (*UBIQUITIN CARRIER PROTEIN*) [51], and *GRMZM2G 153541* (*ELONGATION FACTOR 1A*) [52], were retrieved from the MaizeGDB website. Probe design was done by Bio-Rad specialists and PrimePCR Probe Custom Assay (Bio-Rad) were synthesized accordingly; the main features of probes and primers are listed in S8 Table. qPCR experiments were carried out in a CFX96 Real-time PCR System (Bio-Rad) using SsoAd-vanced Universal Probes Supermix (Bio-Rad) and following manufacturer's specifications. Amplification curves were evaluated using the instrument's software (Vs 3.1) and relative gene expression was calculated by applying the ΔCt method [53–55]. Gene expression was estimated by averaging the ΔCt values computed against either *GRMZM2G102471*, *GRMZM2G153541* or the mean of both as references. Expression in the 2015 experiment was evaluated using solely *GRMZM2G153541* as a reference. *ZCN12* expression levels derived from 173 of the 327 lines studied, because the detection assay was impaired by a three-nucleotide deletion in the 5'UTR of *ZCN12* coinciding with the forward primer annealing site (S11 Fig). Nevertheless, these 173 lines distributed across the genetic groups of our panel in similar proportions as the initial 327 lines. This small deletion had no effects on *ZCN12* transcript accumulation as further assayed by qPCR using the QuantiNova SYBR Green PCR Kit (Qiagen) on a subpopulation of lines characterised by the presence/absence of the deletion.

## Mixed-models, BLUEs and heritability calculation

Analyses were performed using the R software version 3.4.3 [56]. Preliminary data exploration (ANOVA) was conducted to assess the effects of sampling day and time, as well as the actual developmental stage at sampling, on transcripts accumulation (S2 Fig). For all traits measured in the field and platform experiments (flowering-related traits and transcript accumulation), models were selected among several mixed-models to calculate genotypic means. Models were fitted with ASReml-R [57] (version 3), including the line and replicate effects as fixed, random spatial effects, spatially correlated errors, and other fixed effects (year for the field experiments; sampling date or sampling period for transcript accumulation in the platform). For each trait measured in the field and platform experiments (flowering-related traits and transcript accumulation), the best mixed-model was selected to estimate the best linear unbiased estimations (BLUEs) of the genotypic means, which were then used in the rest of the analyses. The same model, but considering the line effects as random, was used to estimate variance components, which were used to calculate broad-sense heritability. Narrow-sense heritability was estimated with a model assuming additive SNP effects using the R-package Heritability [58] (version 1.2) and a relatedness matrix as in Millet et al., [59].

## Structural equation modelling

Structural equation modelling is a generalized method for the analysis of covariance relationships and is used to evaluate the fit of data to a priori causal hypotheses about the functioning of a system [60–62]. These multivariate hypotheses are represented as graphical path models. Structural equation modelling then allows the assessment of the degree of fit between the observed and expected covariances structures, which is expressed as a goodness-of-fit $\chi^2$. Here, the aim was to impose a theoretical structure relating the direct and indirect relationships between *ZCN8*, *ZCN12* and *ZCN7* transcripts accumulation and flowering taking into account the results of bivariate correlations. BLUEs from the platform experiment (2014) were used and the assumption of univariate as well as multivariate normality were verified. Network structures tested were constructed as follow: (i) anthesis was placed downstream of genes expression based on previous knowledge (flowering is a consequence of florigens expression), (ii) then testing for all possible network structures between the 3 florigens acting upstream of anthesis. Models tested are detailed in the results section. Structural equation models were tested in R using the lavaan structural equation modelling package [63] (version 0.6–5), which uses the standard maximum likelihood estimator. A significant goodness-of-fit $\chi^2$ statistic indicates that the model does not fit the data. Once a model has not been rejected and considered biologically plausible, parameter estimates can be used to study direct, as well as indirect, effects of the variables. Standardized path coefficients quantify the strength of a relationship, whereas the effects of the other variables are held constant. Parameter estimates are tested for significance using z statistics. Root mean square error approximation (RMSEA) and comparative fit index (CFI) indices are used to assess the closeness of fit. Good models have a RMSEA < 0.05 and CFI > 0.95.

## GWAS analysis

GWAS was performed on individual traits for each experiment as previously described [59]. Briefly, we used the single locus mixed model:

$$Y = \mu + X\beta + G + E$$

where $Y$ is the vector of phenotypic values, $\mu$ the overall mean, $X$ is the vector of SNP scores, $\beta$ is the additive effect, and $G$ and $E$ represent random polygenic and residual effects. As in Rincent et al. [64], the variance-covariance matrix of $G$ was determined by a genetic relatedness (or kinship) matrix, derived from all SNPs except those on the chromosome containing the SNP being tested. The SNP effects β were estimated by generalized least squares, and their significance ($H_0$: β = 0) tested with an F-statistic. This model accounting for relatedness was found the best to control for confounding factors as compared to models accounting for structure or relatedness plus structure. Analyses were performed with FaST-LMM v2.07 [65]. Physical positions of significant SNPs were projected on the consensus genetic map for Dent genetic material [66]. Candidate SNPs distant less than 0.1 cM were considered as belonging to a common QTL, described via the most significant SNP in the QTL and the interval between all SNPs belonging to the QTL. Co-localizations between QTLs (across traits and experiments) were identified using the same 0.1cM window.

## Regional plot definition

Annotated genes located within QTLs were identified according to maize annotation version 2 (MaizeGDB). For the entire QTL, the pairwise LD ($R^2$) between all variants was estimated using the r-package snpStat [67] (version 1.33.0) and plotted with LDheatmap [68] (version

0.99–7) here shown as a triangle plot. Based on the LD pattern and the levels of significance of SNPs, the QTL were split into sub-regions.

## Exploration of selective pressure for flowering-related traits and transcripts accumulation

Allelic distributions within the population were explored at SNPs representative of the core-set of QTLs and revealed strong contrasts across genetic groups. An $F_{ST}$ study was then undertaken for identifying outlier loci that would differentiate the genetic groups more than the average genomic background. Bayescan version 2.1 [69] decomposes the Locus-population $F_{ST}$ into a component specific for the population shared by all loci (beta) and a component specific for the locus shared by all the populations (alpha). Departure from neutrality at any given locus is assumed when alpha is significantly different from zero. Decisions about the chance of each locus being under selection are made based on posterior odds (PO) comparing the models with and without alpha component. We used as evidence of SNPs under selection markers with $\log_{10}(PO) > 0.5$ which corresponds to markers considered substantially significant [27].

## Re-analysis of RNAseq data

Expression counts relative to RNA-seq experiments and flowering time [35] were downloaded from the Cyverse Discovery Environment under the directory: https://datacommons.cyverse. org/browse/iplant/home/shared/commons_repo/curated/Kremling_Nature3RNASeq282_ March2018. We used data for post-sexual maturity leaves during the day (LMAD) because expression counts for *ZCN7/8/12* at earlier stages (e.g. leaf 3) were extremely low and not available for all lines. We included transcripts accumulation dataset for 204 dent lines (with no biological replicates), of which 182 had also data available for anthesis (BLUPs). We discarded extreme outliers (respectively *ZCN8* > 400, *ZCN12* > 350 and *ZCN7* > 150 expression counts). We thus obtained correlations between the expression counts of each annotated gene in the maize genome and *ZCN8* was calculated (correlation coefficient (r) and associated p-value). The same method was applied to establish the relationship between the expression of each gene and anthesis data. Co-expression data were further reduced by applying a Bonferroni correction and analysed with a k-means clustering approach [70].

## Gene cloning, transformation and growth of *Arabidopsis*

*ZCN12* and *ZCN8* were amplified from genomic DNA from B73 with oligonucleotides lcm116, 117, 118 and 119 as detailed in S7 Table. These primers incorporate sequences for Gateway-based cloning (Invitrogen) and include start and stop codons. PCR products were subject to a second round of PCR amplification with adapters art77 and art78. Gel-purified PCR products were cloned into Gateway pDONR221 vector through BP-mediated recombination and sequenced. *ZCN12* and *ZCN8* entry clones were finally recombined into the binary destination vector pEarleyGate201 [71] via LR clonase. We used the floral dip method to transform wild-type Columbia Arabidopsis and primary transformants were selected on soil by continuous applications of Basta. T1 plants that yielded enough seeds were analysed in T2 to monitor transgene-derived expression by RT-PCR. Plants were grown in a climatic chamber under long-day conditions (16 hours of light, 8 hours of dark), a mean daily temperature of 23 degrees (night-day) and 60% humidity, light was provided by fluorescent tubes and PAR (photosynthetically active radiation) was approximately 90 µmol m$^{-2}$ s$^{-1}$.

## Supporting information

**S1 Fig. A wide flowering time variability conserved across different environments.** (A) Genotypic values of anthesis time, silking and total number of leaves per genetic group, during the platform experiment in 2014. Genetic groups are represented by the name of their founder. D06: n = 14; PH207: n = 42; diverse dents (d.d.): n = 128; Oh43: n = 16; Mo17: n = 35; B14a: n = 41; B73: n = 37. (B) Relationship between genotypic values of the flowering traits (respectively anthesis, silking and total number of leaves) in platform experiments conducted over two consecutive years (2014 and 2015). n = 273 lines in common between both years. (C) Relationship between genotypic values for silking (left panel) and the total number of leaves (right panel) measured in the platform (2014) and field (2015–2016). n = 313 lines. In (A), (B) and (C), genotypic values are BLUEs. Correlation coefficients are displayed together with their significance (*** $p < 10^{-4}$).
(TIF)

**S2 Fig. *ZCN8* transcript accumulation is not consistently affected by day of sampling, time of sampling within a day, or actual leaf stage at sampling.** (A, C) *ZCN8* transcript accumulation over three consecutive days per sub-period of ca. 30 minutes (covering 1.5 to 3 hours after dawn within each sampling day) at stage ligulated leaf 7 (A) or 4 (C). The number of individuals that were sampled is indicated for each day together with the time window of sampling. (B, D) Distribution of the actual number of ligulated leaves within the population at the sampling stage ligulated leaf 7 (B) or 4, (D). Insets show the absence of correlation between *ZCN8* transcript accumulation and the actual number of ligulated leaves.
(TIF)

**S3 Fig. Three methods of normalization of *ZCN8* transcript accumulation show highly consistent results.** (A, B) Correlation between anthesis and expression of *ZCN8* transcript accumulation—respectively measured at stage leaf 7 (A) or leaf 4 (B)—normalized either by *EF1a*, by *UCe*, or by the average ("ave") of *EF1a* and *UCe* transcript accumulations, for individual data (n = 980 plants). (C, D) Correlation between anthesis and expression of *ZCN8* transcript accumulation—respectively measured at stage leaf 7 (C) or leaf 4 (D)—normalized against *EF1a*, *UCe*, or the average ("ave") of *EF1a* and *UCe* transcripts accumulation, for genotypic values (n = 320 lines). In each panel, the X and Y axes show the units of the variable displayed in the main diagonal. Transcript accumulation is expressed as ΔCt. Anthesis time expressed in equivalent days at 20˚C after germination Pearson's correlation coefficients are indicated with their significance (***, $p < 10^{-4}$).
(TIF)

**S4 Fig. *ZCN8* accumulation increases between stage ligulated leaves 4 and 7 in a genotype-dependent fashion.** (A, B) Genotypic values of *ZCN8* accumulation at stages ligulated leaf 4 and 7 in the platform experiment of 2014. *n* = 320 lines. The effect of stage (L4 vs L7) on *ZCN8* accumulation was found significant (p < 0.001) by ANOVA. (C) Relationship between the residuals extracted from the regression between *ZCN8* at stage ligulated leaf 4 vs. 7 displayed in (A), and anthesis time measured in the same platform experiment. Genotypic values are BLUEs. Pearson's correlation coefficients are indicated with their significance (***, $p < 10^{-4}$).
(TIF)

**S5 Fig. Strong correlation between *ZCN8* accumulation and flowering time is confirmed across different environments.** (A) Relationship between genotypic values of *ZCN8* accumulation measured at stage ligulated leaf 7 in the platform in 2014 and the total number of leaves respectively measured in the same experiment (left panel) or the field in 2015–2016 (right

panel). *n* = 313 lines. (B) Relationship between genotypic values of *ZCN8* accumulation in the platform experiment of 2015 on a subset of lines (*n* = 273 lines) and flowering traits respectively measured during the same experiment (left panel, anthesis; middle panel, total number of leaves) or in the field in 2015–2016 (anthesis, right panel). (C) Relationship between genotypic values of *ZCN8* accumulation in platform experiments of 2014 and 2015. *n* = 273 lines. (D) Relationship between genotypic values of *ZCN8* accumulation measured at stage ligulated leaf 4 in the platform (2014) and flowering traits either measured in the same experiment (left panel, anthesis; middle panel, total number of leaves) or the field (right panel, anthesis). *n* = 313 lines. Genotypic values are BLUEs. Pearson's correlation coefficients are indicated with their significance (***, p $< 10^{-4}$).
(TIF)

**S6 Fig. Strong correlation between *ZCN12* accumulation and maize flowering is confirmed across different environments whereas *ZCN7* and *ZmCONZ1* have a weak or no effect.** (A) Relationship between genotypic values of *ZCN12* accumulation measured at stage ligulated leaf 7 in the platform in 2014 and flowering traits measured in the same experiment (total number of leaves, left panel) or the field in 2015–2016 (total number of leaves and anthesis, middle and right panels). *n* = 173 lines. (B) Relationship between genotypic values of *ZCN12* accumulation in the platform experiment of 2015 on a subset of lines (*n* = 160 lines) and, respectively, accumulation of *ZCN8* measured during the same experiment (left panel) or accumulation of *ZCN12* measured in the first platform experiment (right panel). (C) Relationship between genotypic values of *ZCN7* and anthesis time (left panel) and *ZCN8* accumulation (right panel) in the platform in 2014. (D) Relationship between genotypic values of anthesis measured in the platform in 2014 and *ZmCONZ1* accumulation. Genotypic values are BLUEs. Pearson's correlation coefficients are indicated with their significance (***, p $< 10^{-4}$). The red triangle represents genotypic values for the reference line B73.
(TIF)

**S7 Fig. Path diagrams indicate possible direct routes of *ZCN12* to activate flowering.** Models in which *ZCN12* has an indirect effect on anthesis (A) were rejected, whereas models in which both *ZCN8* and *ZCN12* directly affect anthesis were accepted, with an indirect effect of *ZCN7* (B and C). Arrows represent linear functional relationships between anthesis and florigen expression. Simple-headed arrows represent causal relationships, and double-headed arrows represent free correlations. Standardized path coefficients are indicated on each arrow with their level of significance (***: P $< 0.001$, ns: not significant). All three models were tested against our data on BLUEs and results are given in the upper part of each panel. See also Fig 3.
(TIF)

**S8 Fig. *ZCN* eQTLs are major contributors to the genetic variance of flowering time in the platform experiment of 2014.** (A) Number of QTLs shared between anthesis in the platform experiment (2014) and *ZCN* eQTLs. Anthesis was recorded in the platform in 2014 and *ZCN8/7/12* levels were measured in the platform, in 2014 and 2015 at different leaf stages. GWAS was conducted for every trait/experiment, and significant SNPs were grouped according to genetic distances, with a threshold at 0.1 cM to define QTLs. The number of these QTLs is italicised within brackets outside of VENN diagram. The number of common QTLs between traits is italicised within brackets inside each VENN area. QTLs detected were then gathered into 'meta-QTLs' regions containing overlapping individual QTLs. The number of common 'meta-QTLs' is shown in plain text outside and inside the circles, and a common 'meta-QTL' is defined as regions containing at least one QTL of each trait considered. (B) Regional association plot for one meta QTL on chromosome 8. Distribution of the -log$_{10}$(pval) for all variants

in the region. The dotted grey line corresponds to $-\log_{10}$(pval) = 4.5 for anthesis-related traits evaluated in the field (purple) and platform (orange) and 4 for the expression traits evaluated in the platform (green). Bottom panel, linkage disequilibrium (LD) heat map of all SNPs in the QTL showing the local LD ($r^2$) between all the variants; Black lines represent the distribution of the SNPs and the blue boxes represents the genes mapped for the region. Red asterisks represent the position of the most significant SNPs. (See also S4 Table for candidate genes). Grey areas represent sub-regions harbouring the most significant SNPs and genes and having low LD between them on average.
(TIF)

**S9 Fig. Strong patterns of co-regulation are verified in independent datasets.** (A to C) Reanalysis of publicly available data profiling whole genome expression variations from different maize varieties. Comparisons between anthesis and accumulation of respectively *ZCN8* (A, $n$ = 182 lines), *ZCN12* (B, $n$ = 180 lines) (B), and *ZCN7* (C, $n$ = 178 lines) (C). (D-F) Pairwise comparisons between *ZCNs* transcripts accumulation, respectively *ZCN8* vs *ZCN12* (D, $n$ = 200), *ZCN8* vs *ZCN7* (E, $n$ = 198), and *ZCN12* vs *ZCN7* (F, $n$ = 198). Genotypic values (BLUPs) are displayed for anthesis, whereas individual values (no biological replicates) were reported for *ZCN8*, *ZCN12* and *ZCN7* accumulation. Pearson's correlation coefficients are indicated with their significance (***, p $< 10^{-4}$). (G) k-means cluster analysis on the 549 *ZCN8*-coregulated genes based on their degree of correlation with both *ZCN8* expression and flowering time.
(TIF)

**S10 Fig. *ZCN8* and *ZCN12* accumulation follows the flowering time precocity of different admixture groups. (A-C)** Biplots of genetic group values for anthesis vs *ZCN8* accumulation (A), anthesis vs *ZCN12* accumulation (B), and *ZCN8* vs *ZCN12* accumulation (C). Means and standard deviations of genetic group values are calculated from the BLUEs in the platform experiment of 2014. Genetic groups are represented by the name of their founder and the colour code reflects flowering time phenology as in Fig 5A. D06: n = 14; PH207: n = 42; diverse dent: n = 128; Oh43: n = 16; Mo17: n = 35; B14a: n = 41; B73: n = 37. (D-F) Within-group relationship between anthesis and accumulation of respectively *ZCN8* (D), *ZCN12* (E) and accumulation of *ZCN8* vs *ZCN12* (F).
(TIF)

**S11 Fig. A polymorphism in the *ZCN12* region affects amplification efficiency.** (A) Sequence alignment of the *ZCN12* region targeted for amplification in different maize lines. Nucleotides highlighted in grey boxes correspond to forward and reverse primers, respectively whereas the green box corresponds to the probe. Nucleotides highlighted in yellow indicate polymorphisms. Marker AX-90845493 was used to differentiate the B73 from the alternative haplotype in our lines. (B) *ZCN12* Taqman assay using genomic DNA derived from lines carrying the B73 allele (black) or the alternative allele (green). *UCe* amplification was used for normalization and to derive—ΔCt values. (C) *ZCN8* transcript amplification performed on a reduced panel and with a different amplification assay.
(TIF)

**S12 Fig. Graphical abstract of the experiments conducted in this study and of the associated datasets.** Phenotypic data at the plot level in each field experiment (1a) and at the genotypic level in each field experiment and for both field experiment together (1b). Transcripts data at the plant level in each platform experiment (2a) and at the genotypic level in each platform experiment (2b). Phenotypic data at the plant level in each platform experiment (3a) and at the genotypic level in each platform experiment (3b). Description of the genotypic material

used (4). The maize 3D canopy is adapted from Pradal et al., [72]. The maize 3D plant is adapted from Fournier and Andrieu [73].
(TIF)

**S1 Table. Statistical indicators of the genotypic variability for flowering related traits and florigens transcripts accumulation in individual experiments.** Genotypic and residual variance components were estimated with the mixed-model fitted per experiment accounting for fixed replicate effect, random variety and spatial effects and spatially correlated errors. They were used to estimate an interval of broad-sense heritability such as $H^2_{low}$ = varG/(varG + varRes/n) and $H^2_{high}$ = varG/(varG + varRes/n + varSpatial), with varG the genotypic variance, varRes the residual variance and varSpatial the spatial variance. Mean values were calculated using BLUEs estimated with the same mixed model but with variety in fixed effect. The same BLUEs were used to estimate narrow sense heritability with a model assuming additive SNP-effects. Units are as follow: anthesis and silking in thermal time (days at 20˚C), transcript accumulation in -ΔCt.
(XLSX)

**S2 Table. Complete set of QTLs for flowering related traits and florigens transcript accumulation.** GWAS was performed on individual traits and significant SNPs were grouped according to genetic distances, with a threshold at 0.1 cM to define QTLs. Each line contains the information of one QTL with its chromosome (Chr), bin and region (in physical and genetic position), the SNP with the highest *P*-value (SNP Name), its position (SNP Position), -log10(*p*) and the allelic effect of the allele of the reference line B73. Physical positions are based on the B73 reference genome RefGen_v2 and were projected on the consensus genetic map for Dent genetic material to obtain the genetic position.
(XLSX)

**S3 Table. Set of 'meta-QTLs' for flowering-related traits and florigen transcript accumulation.** 346 main regions containing overlapping individual QTLs were identified from S2 Table and defined as 'meta-QTLs'. Each meta-QTL is defined by its Bin, start position in Mbp, end position in Mbp.
(XLSX)

**S4 Table. Core set of QTLs for flowering related traits and florigens transcript accumulation.** List of QTLs for anthesis, silking and total leaf number that colocalize with eQTLs for *ZCN8* and/or *ZCN7* and/or *ZCN12* accumulation. Each QTL is described by flowering trait, name of the peak SNP, chromosome, bin, coordinate of the peak SNP, physical coordinates (start and end), genetic coordinates (start and end), florigen accumulation trait. For these QTLs, putative candidate genes are proposed, selected among the list of *ZCN8*-coexpressed genes (in red) and of known flowering time genes (in green). The established function according to MaizeGDB or a tentative description inferred from the function of the rice/sorghum/Arabidopsis orthologs are also shown. Overlaps with flowering time regions detected in Li et al., [7] are listed. DA = days to anthesis, DS = days to silking (DS), ASI = anthesis–silking interval (ASI), DACV = coefficient of variation of DA, DSCV = coefficient of variation of DS as in [7].
(XLSX)

**S5 Table. Variance components of the different nested models for anthesis (A, B), silking (C, D) and total number of leaves (E, F) in the platform and the field.** Variance components were estimated with a mixed model with random effects for genotype (G), spatial effects, spatially correlated errors and residual error variances (Res) and a fixed effect for replicate (model

M1). In model M2, the complete set of QTLs of the considered flowering trait (e.g. 43 for anthesis in the platform (A)) was added to the fixed terms. In model M3 (respectively M4 and M5), only those QTLs, among the set of the flowering QTLs, which co-localized with eQTLs for *ZCN8* accumulation (respectively *ZCN12* and *ZCN7*) were added to the fixed terms. In model M6, only those QTLs, among the set of anthesis QTLs, which colocalized with *ZCN8* and/or *ZCN12* and/or *ZCN7* accumulation were added to the fixed terms. The proportion of variance retained by the candidate QTLs was estimated by comparing respectively models M2, M3, M4, M5 and M6 to model M1.
(XLSX)

**S6 Table. List of *ZCN8* coregulated genes and their relation to flowering time.** Correlation coefficients (r) as well as statistical parameters (p-value and FDR), are shown to describe relationships between the level of transcript for each gene and either *ZCN8* expression and or anthesis.
(XLSX)

**S7 Table. Fst analysis for SNPs under selection.** List of significant SNPs under selection ($\log_{10}$ posterior odds (PO) > 0.5) for chromosomes 3 and 8. SNPs were found after comparison of the whole population (pop). Rows highlighted in yellow denote SNPs overlapping with core-QTL intervals (QTL). Rows highlighted in green show SNPs that lie within a region that encompasses a coding sequence in an interval of +/- 5 kb. Prob signifies the posterior probability for the model including selection. $\log_{10}(PO)$, the logarithm of Posterior Odds to base 10 for the model including selection. Alpha is the estimated alpha coefficient indicating the strength and direction of selection. Fst is the coefficient estimated by BayeScanFST. The established function according to MaizeGDB or a tentative description inferred from the function of the rice/sorghum/Arabidopsis orthologs are also shown.
(XLSX)

**S8 Table. Primers and assays information used in this study.**
(XLSX)

## Acknowledgments

We thank Michela Landoni and members of the Conti laboratory for their help with tissue collection, Matteo Chiara and Louise Gourlay (University of Milan) for respectively, advice on RNAseq analysis, and critical reading; teams of INRAe Montpellier LEPSE and Saint Martin de Hinx experimental farm for collection of phenotypic data. Alain Charcosset and Cyril Bauland, (INRAe Le Moulon) for the assembly of the population and valuable discussions, Delphine Madur, Valérie Combes and Sandra Negro (INRAe Le Moulon) for genotyping and diversity analysis; we are grateful to David Pot (CIRAD) for valuable suggestions on Bayescan and Denis Vile (INRAe Montpellier) for help with SEM. We also thank partners from companion projects at CREA (Italy), CRB Maize (France), CIAM (Spain), CSIC (Spain), MTA ATK (Hungary), NCRPIS (USA), Univ. of Hohenheim (Germany) who contributed genetic material.

## Author Contributions

**Conceptualization:** François Tardieu, Claude Welcker, Lucio Conti.

**Data curation:** Sara Castelletti, Aude Coupel-Ledru, Carine Palaffre, Llorenç Cabrera-Bosquet, Claude Welcker.

**Formal analysis:** Sara Castelletti, Aude Coupel-Ledru, Italo Granato, Stéphane D. Nicolas, Claude Welcker.

**Funding acquisition:** François Tardieu, Lucio Conti.

**Investigation:** Carine Palaffre, Llorenç Cabrera-Bosquet, Chiara Tonelli, Lucio Conti.

**Methodology:** Stéphane D. Nicolas, Claude Welcker, Lucio Conti.

**Project administration:** Lucio Conti.

**Resources:** Carine Palaffre, Stéphane D. Nicolas.

**Supervision:** François Tardieu, Claude Welcker, Lucio Conti.

**Visualization:** Aude Coupel-Ledru, Italo Granato.

**Writing – original draft:** Sara Castelletti, Aude Coupel-Ledru, Lucio Conti.

**Writing – review & editing:** Sara Castelletti, Aude Coupel-Ledru, Chiara Tonelli, Stéphane D. Nicolas, François Tardieu, Claude Welcker, Lucio Conti.

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
