## [Decision Letter · Decision Letter 0]

29 Oct 2019

Dear Dr Conti,

Thank you very much for submitting your Research Article entitled 'ZCN8 and ZCN12 florigens accumulation in temperate maize explain flowering time variations independent of photoperiod' to PLOS Genetics. Your manuscript was fully evaluated at the editorial level and by three independent peer reviewers. The reviewers appreciated the attention to an important problem, but raised some substantial concerns about the current manuscript. Based on the reviews, we will not be able to accept this version of the manuscript, but we would be willing to review a much-revised version. We cannot, of course, promise publication at that time.

If you decide to revise the manuscript for further consideration at PLOS Genetics, please aim to resubmit within the next 60 days, unless it will take extra time to address the concerns of the reviewers, in which case we would appreciate an expected resubmission date by email to plosgenetics@plos.org.

[LINK]

We are sorry that we cannot be more positive about your manuscript at this stage. Please do not hesitate to contact us if you have any concerns or questions.

Yours sincerely,

Juliette de Meaux

Associate Editor

PLOS Genetics

Gregory Copenhaver

Editor-in-Chief

PLOS Genetics

Comments from the Editors:

Your manuscript has been reviewed by three experts in the field of maize genetics. All three found the results of this study, which associates ZCN12 expression with flowering time and investigates its covariation with ZCN8 and ZCN7 novel and convincing. They have however a number of important comments and criticism on the manuscript. We regret that we cannot accept your manuscript at present but look forward to examine again a carefully revised version.

All comments of the reviewers must be carefully addressed, and the result section must be reorganized and clarified. We note that the text (including abstract and title) contains numerous English grammar mistakes. We ask that you either have the manuscript thoroughly proof-read by a native English speaking colleague, or that you use a professional editing service. Please include a description of how you have addressed this issue in your response-to-reviews letter, along with a detailed description of how you respond to the comments of the reviewers. Furthermore, at least one reviewer noted that not all of the raw underlying data supporting the figures has been provided. In accordance with PLOS Genetics' Open Data policy we ask that you be sure to include it in your revision.

Reviewer's Responses to Questions

**Comments to the Authors:**

Reviewer #1: Review uploaded as attachment

Reviewer #2: The paper by Castelletti et al combines diversity panel analysis using a wide range of maize inbreds and large scale expression analysis to further our understanding of the genes that control the floral transition. In particular they examine expression in leaf tissue and therefore are focused on long distance signalling molecules, or florigens. Only the ZCN8 gene has been identified as a veritable maize florigen, however many plant species reportedly have multiple florigen genes. The focus here is on ZCN8 and a putative florigen, ZCN12. The researchers use expression profiling and transgenic analysis to show that ZCN12 encodes a florigen and, more relevant to this study, ZCN8 and 12 cooperate to control flowering in temperate maize, which is essentially day-neutral and therefore not dependent on photoperiod-induced flowering (i.e. autonomous). Further, they suggest that ZCN7, a putative syntenic paralog of ZCN8, may act upstream of both ZCN8 and 12 to cause flowering. A better understanding of the elements underlying flowering in autonomous maize is welcome, as this is a particularly intractable problem and not s easily addressed as has been done in photoperiod dependent plants.

Overall the paper is well written and the ideas behind the analysis are laid out fairly clearly (with the exceptions noted below). There is perhaps a sight over-interpretation of some of the data, in particular with respect to the model building based on expression analysis. In fact the overarching conclusions regarding how these genes may cooperate to control flowering is highly dependent on speculative modeling, and therefore the underlying conclusions are not as robust as they could be. Even so, given the complexity of the maize genome and the rapid evolution of maize from photoperiod dependent to autonomous flowering, the results presented here at least provide some inroads for further testing to the proposed models.

Some specific points:

1. Expression changes in the ZCN8 gene in autonomous maize are reported to be fairly constant, although it accumulates as the plants age. The authors take great care in explaining the experimental design, which is appreciated because too many studies do not take this into account.

2. The diversity panel examined here no doubt contains maize from a fairly large geographic range. Previous studies (e.g. Cole et al 2010) show that different maize lines exhibit a range of photoperiod responses (i.e., only a few northern lines are truly day-neutral). Therefore, So the question is, does this have an impact on the interpretation of the data? That is, it seems the more photoperiod-sensitive maize is, the greater the expression changes in florigen genes (as was shown for ZCN8). If so, do the authors need to take this into consideration when proposing models? The current model assumes “desensitization to photoperiodic cures” (lines 38-39), but they should acknowledge that maize flowering is not “either one or the other”.

3. This study focuses on 3 florigen genes 7,8 and 12), one of which is verified here and elsewhere. But are these the only one and should they take into consideration other possible florigens? They might want to mention the findings of Stephenson et al (2019) where they report that ZCN14, 15 and 18 have florigen activity as well. Of course this would increase the complexity of the model, but is there a reason to not include them?

4. They says ZCN8 expression is negatively correlated with flowering time. Although technically correct, this is confusing and not normally used. True, as ZCN8 expression increases the days to flowering decreases, but nonetheless it would be easier to understand if they state clearly that ZCN8 expression has a positive effect on flowering.

5. Lines 107-108: The authors state that ZCN8 decrease causes a strong delay in flowering, as reported by others. But this is an overstatement. The only report, as far as I know, of ZCN8 knockdown was by Meng et al (2011), and they found only slight delay in flowering (3 or 4 leaves). Whereas mutants such as dlf1 and id1 produce many more leaves and show a true strong effect. Also of the cited papers indicating a strong effect on flowering, only Meng et al, 2019 show these data; the other papers, (Stephenson et al 2019 and Liang et al, 2019) do not as far as I can tell.

6. The co-regulation of ZCN8 and 12 and how they might interact is one of the novel findings of this paper. This is partly based on the SEM analysis presented in Figure 3. I have to confess that I do not understand how this model was built. Although explained in the Methods section, I had a hard time understanding how the pathways were tested and why A and B were rejected. Perhaps those well-versed in this type of modeling have a better understanding of this type of analysis, but I think others would benefit from a more lucid explanation of how this was put together and what it means “in the real world” of floral induction.

(by the way, line 246: “synergic” should be “synergistic”)

7. Given the predicted redundant roles played by ZCN8 and 12, it would be interesting to see whether a double knock out is very late flowering. Not a trivial experiment in maize, but worth thinking about.

Reviewer #3: The manuscript by Castelletti et al deals with florigene (ZCN8) and florigene-like/related gene expression across a large maize panel. The objective was to investigate how such florigenes are expressed and regulated in temperate (= mostly photoperiod insensitive) maize materials. Wild maize (= teosinte), and most maize landraces are photoperiod sensitive, thus they are strongly flowering-delayed when grown in temperate latitudes, while modern temperate maize cultivars or hybrids appear mostly photoperiod insensitive. The expansion of maize from tropical latitudes to northern or southern environments must have required dramatic changes in photoperiod sensitivity and flowering time, however how these changes involved florigene(s) expression changes is indeed not understood yet.

The topic is of substantial scientific and practical (breeding) interest. The experimental strategy deployed by the authors was in my opinion essentially correct and potentially informative. The technical tools applied were appropriate. It is worth mentioning that phenotypic data and leaf samples for gene expression analysis were collected in a large, well replicated experiment which was carried out in a state-of-the-art phenomics facility.

Main results

A. identification of a large range of variation of ZCN8 gene expression and strong correlation between ZN8 and flowering time in temperate maize materials, which suggested that flowering time variation involves variation of ZCN expression.

B. Similar results for ZCN12, which was also shown to act as flowering inducer in Arabidopsis when overexpressed.

C. A strong coincidence between ZCN8 and ZCN12 eQTLs and flowering time QTL.

D. Lack of detection of trends of co-expression between ZCNs and genes corresponding to eQTL for ZCNs (Results lines 327-329; Discussion lines 448-451).

Comments.

1. A major problem for this manuscript is the large number of experiments and thus the consequent necessity for the authors to present and discuss many results. I have to admit that, for the RESULTS section, it was not always easy to follow the experiments succession and the underlying rationale. Therefore I would strongly suggest the authors to subdivide RESULTS paragraphs in shorter ones, corresponding to single experiments and/or single hypothesis under testing. In my opinion, text length can be safely reduced. A CONCLUSIONS section would probably help too.

2. Some RESULTS parts could go either in MAT and MET of in DISCUSSION. For instance, lines c. 128-132 and 160-166 reported technical details typical of MAT and MET. Lines 193-195 can probably be deleted. Line 198-199 seem to be a contradiction of the overall message of this message, so please check and perhaps delete.

3. Concerning point D above, I wonder whether strong post-transcriptional regulation (eg. by miRNA) or post translational regulation (at protein level) could be listed as possible causes behind lack of correlation. At least for ZmRap2.7, it was clearly shown that besides being regulated by a long distance regulatory element, it was also post-transcriptionally regulated by miR172.

4. Fig. 4 and related text. I suggest to compute and show LD relationships for the whole region (ie. from left hand side to right end side), and not just for three somewhat arbitrarily chosen regions. Long range LD relationships would probably prevent drawing any conclusions about the possibility to locate a single causative variants, or any other general trends. Additionally, if appropriate, the authors should recognize that long LD at this locus was already observed previously using a different maize panel (Hirsch et al. 2014, cited in this manuscript as [24]).

5. Maize should be maize (no capital letter).

6. QTL or QTLs (in case of multiple loci)? There is no consistency across manuscript.

7. I have the impression the title might contain mistakes. Please consider that ‘explain’ could be ‘explains’ and ‘variations’ could be ‘variation’.

**Have all data underlying the figures and results presented in the manuscript been provided?**

Reviewer #1: No: None of the original phenotypes or genotypic data is provided. Only processed data and results

Reviewer #2: Yes

Reviewer #3: Yes

PLOS authors have the option to publish the peer review history of their article (what does this mean?). If published, this will include your full peer review and any attached files.

Reviewer #1: No

Reviewer #2: No

Reviewer #3: No

---

## [Decision Letter · Decision Letter 1]

22 May 2020

Dear Dr Conti,

We are pleased to inform you that your manuscript entitled "Maize adaptation across temperate climates was obtained via expression of two florigen genes" has been editorially accepted for publication in PLOS Genetics. Congratulations!

Before your submission can be formally accepted and sent to production you will need to complete our formatting changes, which you will receive in a follow up email.

Please be aware that it may take several days for you to receive this email; during this time no action is required by you. Please note: the accept date on your published article will reflect the date of this provisional accept, but your manuscript will not be scheduled for publication until the required changes have been made. You MUST make sure you upload the correct version of the manuscript, though. There was a mistake in the previous submission.

Yours sincerely,

Juliette de Meaux

Associate Editor

PLOS Genetics

Gregory P. Copenhaver

Editor-in-Chief

PLOS Genetics

Comments from the reviewers (if applicable):

Dear Authors,

Your manuscript has been revised in depth and addressed all the criticisms raised by the reviewers in the previous round of examination. Due to the covid crisis, we have not been able to provide a faster turnover and we apologize for that. Note also that additional delay was caused by the fact that the manuscript you submitted is the previous version. The final version could nevertheless be evaluated thanks to the file with highlighted changes. This was not optimal. Anyways, I find the final version now more succinct and undoubtedly clearer. I believe it is suitable for publication and will make a significant contribution to the field. Now, please make sure that you submit the correct version prior to proof preparation, so that there is no major error at the stage of manuscript preparation.

Reviewer's Responses to Questions

**Comments to the Authors:**

Reviewer #3: Authors have satisfactorily addressed all my comments and suggestions.

**Have all data underlying the figures and results presented in the manuscript been provided?**

Reviewer #3: Yes

PLOS authors have the option to publish the peer review history of their article (what does this mean?). If published, this will include your full peer review and any attached files.

Reviewer #3: No

**Data Deposition**

http://datadryad.org/submit?journalID=pgenetics&manu=PGENETICS-D-19-01500R1

**Press Queries**

---

## [Editor Report · Acceptance letter]

6 Jul 2020

PGENETICS-D-19-01500R1 

Maize adaptation across temperate climates was obtained via expression of two florigen genes 

Dear Dr Conti, 

We are pleased to inform you that your manuscript entitled "Maize adaptation across temperate climates was obtained via expression of two florigen genes" has been formally accepted for publication in PLOS Genetics! Your manuscript is now with our production department and you will be notified of the publication date in due course.

With kind regards,

Kaitlin Butler

PLOS Genetics

On behalf of:
